# A Chemoptogenetic Tool for Spatiotemporal Induction of Oxidative DNA Lesions *In Vivo*

**DOI:** 10.3390/genes14020485

**Published:** 2023-02-14

**Authors:** Suhao Han, Austin Sims, Anthony Aceto, Brigitte F. Schmidt, Marcel P. Bruchez, Aditi U. Gurkar

**Affiliations:** 1Aging Institute of UPMC, The University of Pittsburgh School of Medicine, 100 Technology Dr, Pittsburgh, PA 15219, USA; 2Molecular Biosensor and Imaging Center, Carnegie Mellon University, 4400 Fifth Avenue, Pittsburgh, PA 15213, USA; 3Division of Geriatric Medicine, Department of Medicine, University of Pittsburgh School of Medicine, 3471 Fifth Avenue, Kaufmann Medical Building Suite 500, Pittsburgh, PA 15213, USA

**Keywords:** ROS, aging, DNA damage, optogenetic

## Abstract

Oxidative nuclear DNA damage increases in all tissues with age in multiple animal models, as well as in humans. However, the increase in DNA oxidation varies from tissue to tissue, suggesting that certain cells/tissues may be more vulnerable to DNA damage than others. The lack of a tool that can control dosage and spatiotemporal induction of oxidative DNA damage, which accumulates with age, has severely limited our ability to understand how DNA damage drives aging and age-related diseases. To overcome this, here we developed a chemoptogenetic tool that produces 8-oxoguanine (8-oxoG) at DNA in a whole organism, *Caenorhabditis elegans*. This tool uses di-iodinated malachite green (MG-2I) photosensitizer dye that generates singlet oxygen, ^1^O_2_, upon fluorogen activating peptide (FAP) binding and excitation with far-red light. Using our chemoptogenetic tool, we are able to control generation of singlet oxygen ubiquitously or in a tissue-specific manner, including in neurons and muscle cells. To induce oxidative DNA damage, we targeted our chemoptogenetic tool to histone, *his-72,* that is expressed in all cell types. Our results show that a single exposure to dye and light is able to induce DNA damage, promote embryonic lethality, lead to developmental delay, and significantly reduce lifespan. Our chemoptogenetic tool will now allow us to assess the cell autonomous versus non-cell autonomous role of DNA damage in aging, at an organismal level.

## 1. Introduction

Reactive oxygen species (ROS) are produced continually through life and have physiological and pathological ramifications. ROS can cause oxidative damage to macromolecules including DNA, RNA, lipids, and proteins, but is also known to play a critical role in signaling cascades [1,2]. The oxidative stress theory of aging has been controversial, with studies that have shown both beneficial and detrimental effects of ROS [2,3,4]. Stemming from multiple studies, current view posits that the level of ROS, as well as the spatial and temporal context, is critical when understanding the role of ROS.

Exposure to exogenous and endogenously produced ROS drives oxidative DNA damage and is a challenge to genome stability [5,6]. Oxidative DNA damage is highly variable between cell types and tissues, not only in absolute lesion numbers but also distribution over the genome [7,8,9,10,11,12,13]. Oxidative DNA damage is associated with aging and multiple age-related diseases, including neurodegeneration and cancer. Chemotherapeutic agents used to treat cancer patients cause oxidative DNA damage and contribute to long-term side effects including premature aging syndromes such as frailty [14,15]. It is well-documented that oxidative nuclear DNA damage increases in all tissues with age [16,17]. However, delineating the biological impact of oxidative DNA lesions has been challenging since oxidants used to cause DNA damage also affect redox status and damage other macromolecules. Moreover, it is unclear whether systemic application of ROS generators such as hydrogen peroxide, paraquat, etc. recapitulates endogenous ROS signaling and detoxification mechanisms. 

A more precise understanding of when and where oxidative DNA damage that is destined to accelerate age-related pathologies arise, and how it drives aging are long-standing questions in the field. For this challenge, we need to locally induce ROS, limit its diffusion, and control its production in a spatiotemporal manner. Here, we developed a chemoptogenetic tool that produces primarily 8-oxoguanine (8-oxoG) exclusively at DNA in a whole organism, *C. elegans*. This photosensitizer, fluorogen activating peptide (FAP), has been targeted to telomeres in vitro previously to test the role of oxidative damage at telomeres in proliferating cells [11,18]. Telomeric 8-oxoG promotes telomere fragility, DNA damage response signaling, and premature cellular senescence. We have now adapted this chemoptogenetic tool to induce localized ROS in live animals to track and understand the spatial and temporal role of oxidative DNA damage in driving healthspan and aging, *in vivo*.

## 2. Materials and Methods

### 2.1. C. elegans Strains 

All strains were maintained on nematode growth medium (NGM) plates seeded with *E. coli* OP50 at 20 °C unless overwise noted. Parental strains *bus-5* (br19), N2, and FAS46 (*uge30[gfp::his-72*]) III were obtained from CGC. Independent lines of transgenic worms, either extrachromosomal or integrated, were maintained at least 3 generations and confirmed for normal morphology and behavior before conducting any experiments. The complete list of strains used are listed in Table 1. 

### 2.2. CRISPR Generated FAP::GFP::HIS-72 

To generate strain AUG31 expressing FAP::GFP::HIS-72, we engineered CRISPR-Cas9 mediated genome insertion of *fap* gene into parental strain, FAS46 *(uge30[gfp::his-72*]) III. Briefly, crRNA (TCCAGAACAACAATGAGTAA) designed to target N-terminal of *gfp::his-72* III, tracrRNA and Cas9 protein were ordered from IDT. dsDNA repair template containing *fap* and 35 bp homologous arms on each side were generated by PCR using Donor-*fap*-*his-72*-F: 5′-ccaaccattctcaattctcaatttccagaacaacaatgcaggccgtcgttacccaag-3′, and Donor- *fap*-*his-72*-R: 5′- acaactccagtgaaaagttcttctcctttactcatggagaggacggtcagctggg-3′. In vitro assembled Cas9-crRNA-tracrRNA RNP, melted repair template, and injection marker PRF4::*rol-6 (su1006)* plasmid (a gift from Dr. Lamitina) were mixed and microinjected into strain FAS46 [20]. 

To screen, about 50 individual F1 rollers were isolated onto separate OP50 plates and allowed to have progeny prior to genotyping by PCR. Single F1 rollers were lysed by WLB (10 mM Tris-Cl, pH 8.2, 50 mM KCl, 2.5 mM MgCl_2_, 0.45% Tween 20, 0.05% gelatin) with 1 mg/mL proteinase K at 60 °C for 1 h, and proteinase K was inactivated at 95 °C for 15 min. The insertion site was PCR amplified to see differences in size, and non-roller F2s from positive F1 plates were picked individually and genotyped for homozygotes. Final positive homozygotes were kept several generations and observed no visible morphological or behavior defects. The insertion site including upstream and downstream of *fap::gfp* gene was PCR amplified and confirmed by sequencing. 

### 2.3. MG-Dye Treatments and Red light Illumination 

A whole plate of worms was washed along with OP50 into a glass petri dish in 3 mL M9 buffer. MG-ester (cell permeable dye that does not generate singlet oxygen) or MG-2I were added to a final concentration of 2 μM. The glass plates were shielded from light and placed on a shaker at 100 rpm at room temperature for the desired time. Experimental or control worms were pelleted by brief centrifugation and transferred onto OP50 plates. Red light illumination was performed with a customized LED (660 nm) box that delivers 89 mW/cm^2^ intensity for desired time. 

### 2.4. Behavior Analysis and Quantitation

Complete paralysis in *eft-3p::fap::gfp* worms was defined as unable to move even after gentle touch with a worm pick. Unc and coiled phenotype of *unc-17p::ph::fap::gfp* worms were defined as unable to move properly after touching and coiled body bends. To quantitate the percentage of paralysis, at least three independent experiments (with 30–60 animals each) were performed. 

Pumping rate of *myo-2p::fap::gfp* worms were counted under standard dissecting microscope. To confirm intake of *E. coli*-GFP (a gift from Dr. Arjumand Ghazi), worms were transferred onto NGM plates with *E. coli*-GFP for 30 min and immobilized with sodium azide solution for imaging. 

### 2.5. Immunostaining of Rad-51 Foci 

After 16 hr soaking with MG-2I, FAP::GFP::HIS-72 expressing worms were treated with or without 20 min red light. The worms were placed on OP50 for 2.5 h before dissecting gonads for immunostaining. Immunostaining and analysis of RAD-51 foci [21] was performed following previously established protocol [22]. Briefly, day 1 (D1) adult gonads were dissected in 1X sperm salts (50 mM PIPES pH 7.0, 25 mM KCl, 1 mM MgSO_4_, 45 mM NaCl, and 2 mM CaCl_2_) and fixed in 1% paraformaldehyde diluted in 1X sperm salts for 5 min in a humid chamber. Slides were then frozen on a metal block on dry ice for at least 10 min prior to flicking off the cover slip and immersing in methanol precooled in dry ice for 2 min. This was followed by 5 s in acetone at 4 °C. Slides were washed in PBST with BSA [1XPBS with 0.1% Tween and 0.1% bovine serum albumin (BSA)], and incubated overnight at 4 °C with primary antibody. Antibodies used: rabbit anti-RAD-51 (gift from Dr. Sarit Smolikove) and mouse Mab414 (gift from Dr. Judith Yanowitz). The next day, slides were washed 3× in PBST with BSA for 10 min and incubated with secondary antibody (goat anti-rabbit Alexa 594 and anti-mouse alexa 488, 1:1000 in PBST with BSA) for 2 h at room temperature in the dark. Slides were then washed 2 × 10 min in PBST with BSA. Slides were mounted in Prolong Gold with DAPI and put in the dark to cure overnight before imaging. 

### 2.6. Embryonic Lethality and Developmental Delay

Twenty day 1 adults untreated (UT), with dye (+D) and dye+ light (+DL) were placed on OP50 seeded NGM plates in triplicate. After a one-hour egg lay, 40–60 eggs were transferred to a new plate in triplicates. The worms were grown at 20 °C for 36 h. Three independent experiments were performed. 

Number of unhatched eggs versus total number of eggs was recorded (blinded to experimental group) as % embryonic lethality.

Number of non-adults (YA or larval stage) versus total number of hatched worms was recorded (blinded to experimental group) as development delay.

### 2.7. Lifespan Analysis

Lifespan analysis was performed as described [23], with a few changes. Briefly, 40–60 animals were used per conditions and scored every other day in a blinded manner. Day 1 adults were transferred to 2.5 μg/mL FUDR containing NGM plates and were maintained on FUDR for the rest of the lifespan analysis. All lifespan experiments were performed at 20 °C in a blinded fashion.

### 2.8. Imaging

For fluorescence imaging of FAP::GFP and MG-ester, worms were anesthetized with 10 mM sodium azide on 2% agarose pads. Images were captured on Leica SP8 confocal microscope with 20× or 63× objective. Images were processed by Leica software. Movies were captured on Zeiss SteREO equipped with AxioCam camera. 

### 2.9. Statistics

All statistical analyses were performed with GraphPad Prism 8.0 (GraphPad Software, Inc., San Diego, CA, USA). Survival analyses were performed using the Kaplan–Meier method and the significance of differences between survival curves was calculated using the log rank (Mantel–Cox) test. Differences between groups were considered to be significant at a *p* value of <0.05. One way ANOVA was performed with multiple comparisons for most assays. Detailed information is in Figure Legends.

## 3. Results

### 3.1. Systemic Induction of ROS Promotes Paralysis

We used a genetically encoded ROS generator, a fluorogen activating peptide (FAP) with high affinity for photosensitizer dye, malachite (MG-2I). FAP is a 25 kDa binding peptide for malachite green (MG) derivatives with a low-picomolar dissociation constant [24,25,26]. Excitation with far-red light allows for non-toxic penetration to tissues in live animals. FAP binding and excitation allows MG-2I to generate singlet oxygen, (^1^O_2_), locally (Figure 1A). Note that ^1^O_2_ is highly reactive, with limited diffusion, and cannot be converted into other ROS, thus offering multiple advantages [27]. The FAP chemoptogenetic tool has been used for protein inactivation, targeted cell killing, and rapid targeted lineage ablation *in vivo* in zebrafish. However, the zebrafish model required microinjection [28,29] of the MG-2I dye, restricting its usage for any lifespan or healthspan assays. Therefore, we first wanted to test the penetration of the MG dye *in vivo* in *C. elegans*, since the worm is protected by a cuticle that acts as a barrier to chemical uptake. 

To overcome this challenge, we tested mutants with altered cuticle properties to increase permeability of the MG dye. The mutant, *bus-5,* encodes the dTDP-glucose 4,6-dehydratase, *rml-2*, and is required for dTDP rhamnose biosynthesis and is suggested to be a cuticular component [30]. *bus-5 (br19)* has a compromised cuticle but retains integrity and biological function [31]. First, to test the systemic induction of ROS with our chemoptogenetic tool, we cloned the promoter of a ubiquitously expressed gene *eft-3*, a translation elongation factor 1-alpha homolog, to drive expression of FAP in worms. Here, FAP was tagged with GFP in frame (codon optimized with *C. elegans* introns) to track its expression and regulation by *unc-54* 3′UTR. We expressed this transcriptional reporter *eft-3p::fap::gfp* in *bus-5* worms as an extrachromosomal array. The transgenic animals expressed GFP signal in all tissues throughout their lifespan, as expected (Figure 1B,C). The transgenic worms that express *eft-3p::fap::gfp* showed diffuse green fluorescence in all cell types, except in the germline. Multiple independent lines were maintained for >3 generations before conducting further experiments. All transgenic animals were indistinguishable from *bus-5* parental strains and did not exhibit any obvious morphological or behavior defects.

First, we injected the transgenic lines with MG-ester fluorogen to examine penetration of the dye *in vivo*. As seen in zebrafish, microinjection of MG-ester dye, which binds FAP but does not produce ROS, showed co-localization of red fluorescence (MG-ester) with green fluorescence (FAP-GFP) (Figure 1C). However, injection with MG would limit longitudinal analysis of *C. elegans* for healthspan and lifespan measures. Therefore, we next tested soaking of transgenic worms with MG-ester for multiple time points in different life stages. Interestingly, soaking for 1, 4, and 24 hr showed co-localization of MG-ester signal with GFP, confirming that soaking for a short period allows cell permeability of the MG-ester dye. Additionally, the red fluorescence signal can be detected at all stages of worms, from larval to adults, suggesting intake of MG dye at a variety of life stages (Figure 1C). For the next set of experiments, we used overnight soaking (~14–16 hr) with MG-2I, a di-iodinated derivative that has been previously shown to produce singlet oxygen upon excitation with far-red light (Figure 1A). After illumination using an LED source at 660 nm, the transgenic worms were paralyzed immediately in a light exposure-dependent manner, retained a rigid posture for about 24 hr, and were eventually dead (Figure 1D,E, Appendix A). Neither MG-2I dye treatment or red light illumination alone caused visible toxicity to *bus-5* transgenic lines.

### 3.2. Chemoptogenetic Tool Allows for Temporal Induction of ROS

As we optimized MG delivery and induction with light in the *bus-5* transgenic lines, we next wanted to optimize the system in wildtype (N2) *C. elegans.* We expressed the transcriptional reporter, *eft-3p::fap::gfp* in N2 worms as an extrachromosomal array. Soaking worms overnight (~14–16 hr) in MG-ester allowed for uptake in almost all tissues examined. Similar to *bus-5*, after 2 min illumination with far-red light, 50% of the transgenic animals were paralyzed, and within 5 min 100% of the population was paralyzed (Figure 2A). The N2 extrachromosomal array transgenic lines performed as well as the *bus-5* line.

Since our goal is to use our FAP chemoptogenetic tool to induce oxidative damage through life and examine its role in aging, we wanted to test the temporal induction of ROS using our tool. Previously, it has been reported that FAP has a low-picomolar dissociation constant. Therefore, we wanted to test whether MG could be fed to young animals and be retained in the worm for extended period of time. The *eft-3p::fap::gfp* was integrated into N2 wildtype in order to ensure stable expression throughout lifespan. Strikingly, a single overnight soak at larvae (L4) stage allowed for MG-ester binding and fluorescence at least until day (D9) of adulthood (Figure 2B). Since we expected *eft-3p::fap::gfp* integrated lines to retain low copy FAP, we used 20 min light exposure to examine paralysis. We tested whether light and MG-2I dye at different ages would be effective to induce paralysis as observed in young worms. After soaking with MG-2I overnight and exposure to 20 min red light, D5 and D9 adults paralyzed at the same rate compared to young D1 adults (Figure 2C). Taken together, our data shows that the FAP-chemoptogenetic tool can be used to temporally induce ROS and is suitable to address lifespan and healthspan. 

### 3.3. Chemoptogenetic Tool Allows for Spatial Induction of ROS

Tissue-specific oxidative damage has been challenging to address since we need a tool that can be precise, sensitive, and adaptable to multiple tissues. To explore if the FAP-chemoptogenetic tool can specifically target certain tissues, we chose two tissues: neurons and pharyngeal pump (Figure 3A). Genetically encoded and inducible tools for neuronal ablation have been in high demand to study neurodegeneration. However, there is no optogenetic tool that can be activated by near infrared light (NIF) light, which would allow deeper tissue penetration and cause drastically less damage in free living *C. elegans.*

To test the efficacy of neuron ablation with the FAP-chemoptogenetic system, we generated transgenic *bus-5* worms that express FAP::GFP in cholinergic motor neurons, using the *unc-17* promoter. Previous studies have reported target cell ablation can be achieved through localized ROS production [32,33]. Specifically, plasma membrane targeted ROS leads to lipid peroxidation at the membrane, driving neuronal cell ablation and promoting paralysis [33]. Therefore, the N terminal of FAP::GFP was tagged with a membrane targeting sequence, the PH (Pleckstrin Homology) domain from rat PLC-δ [34]. PH domain binds to phosphoinositides and targets FAP to the plasma membrane. By targeting FAP to the plasma membrane, we first wanted to test targeting singlet oxygen to lipids and promote neuronal cell ablation. Extrachromosomal *unc-17p:ph::fap::gfp* expressing *bus-5* worms showed GFP expression in ventral and dorsal nerve cords and motor commissures (Figure 3B), confirming FAP expression in cholinergic motor neurons. Worms expressing an extrachromosomal version of the transgene did not show any physiological or morphological defects. Soaking with MG-2I or MG-ester did not affect locomotion behavior. To examine neuronal ablation, we synchronized *unc-17p:ph::fap::gfp* transgenic worms at L4 stage and soaked them overnight with MG-2I. After exposure to MG-2I, and immediately after red light illumination, *unc-17p:ph::fap::gfp* transgenic worms were uncoordinated (Unc) and coiled, a phenotype that is correlated with cholinergic motor neuron ablation (Figure 3C). Within 5 min of illumination, transgenic worms began to show various degrees of Unc phenotype. After 20 min of exposure to red light, 100% of the worms displayed Unc phenotype (Figure 3D, Appendix A). Locomotion defects persisted 24 hr after exposure to light, suggesting permanent motor neuron death/ablation. The neuronal damage was confirmed under confocal microscopy (Figure 3B). After 24 hr of red light illumination, dorsal neuron body and commissure was undetected, while the remaining ventral neurons displayed GFP puncta (Figure 3B). 

To test whether we can drive oxidative stress in the muscle, we targeted the pharynx, using the promoter of myosin heavy chain structural gene, *myo-2*. MYO-2 is abundantly expressed in the pharynx muscle and neuronal cells throughout lifespan (Figure 3A). The transcriptional reporter P*myo-2*::FAP::GFP in *bus-5* worms showed strong GFP signal in the pharynx, and normal morphology and behavior under basal conditions (Appendix A). Synchronized L4 worms were soaked with MG-2I and exposed to red light. Within 5 min of illumination, ~100% worms ceased pumping (Figure 3E), although they maintained normal body movement (Appendix A). Pharyngeal function is also required for efficient food intake. The pharyngeal pump consists of coordinated contraction and relaxation required for feeding. Fluorescently labeled bacterial intake can therefore inform of pharyngeal activity. Interestingly, blue light has shown to increase pumping rate and may therefore act as a limitation of other optogenetic tools. We exposed transgenic P*myo-2*::FAP::GFP worms to fluorescently labeled *E. coli* (OP50), and then exposed them to 660 nm red light for 5 min or left them unexposed to light. GFP OP50 was found in the gut of transgenic worms with no exposure to red light, but not after exposure to light and dye (Figure 3F), suggesting that oxidative stress in the pharyngeal muscle inactivates pumping. Taken together, our data shows that the FAP-chemoptogenetic tool allows for spatial induction of oxidative damage in live animals.

### 3.4. Chemoptogenetic Tool Allows for Oxidative DNA Damage

Our goal is to induce oxidative DNA damage in *C. elegans* and examine its role in healthspan and lifespan. Therefore, to test whether our chemoptogenetic tool can induce oxidative DNA damage, we tagged endogenous histone component, HIS-72, with FAP::GFP by CRISPR/Cas9. HIS-72 encodes H3.3-like protein and contains the evolutionarily conserved AAIG motif. HIS-72 is expressed ubiquitously in all tissues over the lifespan of worms, making it suitable for our study. Importantly, *his-72::gfp* or loss of *his-72* did not affect lifespan in worms [35]. GFP::HIS-72 was tagged with FAP (Figure 4A) and confirmed to ubiquitously express. We and others have previously shown that an increase in DNA damage results in embryonic lethality, measured as egg-hatching capacity. To test whether induction of oxidative DNA damage also results in unhatched embryos, we used FAP::GFP::HIS-72 expressing worms. Young adults were left untreated (UT); treated with dye overnight, MG-2I (+D) alone; or treated with dye and 660 nm light (+DL) for 20 min. Immediately after illumination, the adults were allowed to lay eggs for 1 h (F0_1_) or after 24 hr post-exposure (F0_24_).

Exposure to both dye and light significantly compromised egg hatching capacity in the F0_1_ generation (Figure 4B). Compared with control groups (untreated or with dye alone), the average hatching rate decreased by ~50% in all three lines tested. Interestingly, embryonic lethality of the F0_24_ was not affected, suggesting efficient DNA damage resolution and repair. DNA damage is also known to delay larval development, and is a measure to evaluate DNA damage, response, and repair efficiency. Consistent with other studies, oxidative DNA damage delayed the development of larvae from the stage-synchronized F0_1_ population (Figure 4C). We also examined RAD-51 foci in +D and +DL treated animals. RAD-51 is a DNA binding protein and mediator of DNA damage. Upregulated RAD-51 foci are indicative of increased DNA damage or repair defects. Immediately after exposure to dye and light, there was a significant increase in RAD-51 foci in the germline that was resolved with time (Figure 4D,E), thus confirming that our chemoptogenetic tool can induce DNA damage in a targeted manner. Finally, we followed the lifespan of transgenic worms exposed to dye alone or dye + light. Interestingly, induction of oxidative DNA damage led to reduced lifespan, as evidenced in other DNA repair mutants (Figure 4F). Therefore, taken together, these results validate that our chemoptogenetic tool can be used *in vivo* to drive oxidative DNA damage in *C. elegans.*


## 4. Discussion

DNA is the primary template encoding genetic information, but it is constantly bombarded with damaging agents, making it surprisingly unstable [17,36]. DNA repair has been documented to decline with age, whereas nuclear DNA damage correlates with age-related pathologies, including, diabetes, cancer, and neurodegeneration. Thus, there is robust evidence for DNA damage as a major causal contributor to human aging and disease. Elegant studies have generated DNA endonuclease tools that can cause double strand DNA breaks (dsDNA) [37,38]. However, controlling the threshold of damage in such systems has been a challenge. Moreover, dsDNA breaks are rare and toxic (cause apoptosis), whereas oxidative DNA damage are very prevalent. Our results not only validate that our chemoptogenetic tool can cause DNA damage in *C. elegans,* but also confirms that oxidative DNA damage can drive aging.

FlAsH and ReAsH, KillerRed, and MiniSOG are genetically targeted photosensitizers that have been previously developed to produce ROS at specific targets in cells [33,39,40,41,42,43]. These tools have been mainly used for cell ablation. However, some of the challenges with these photosensitizers has been that they require a very high light dose to produce ROS, and the spectral properties of these overlap with biological chromophores. This leads to the challenge of ROS production even in the absence of photosensitizers [44,45]. Similarly, blue light that is used to generate ROS by some of these photosensitizers is perceived as toxic in multiple animals, thus limiting the ability to examine healthspan and lifespan trajectories [45,46,47,48].

We generated a chemoptogenetic tool to directly test whether DNA damage alters aging hallmarks in a cell autonomous or non-cell autonomous manner. This technology offers significant advantages: (1) It acts in a very restricted radius and can be targeted to specific cells/tissues. (2) The fluorogen is only excited when bound to the FAP, which improves the signal to noise ratio. (3) The FAP and dye are not harmful until bound and excited. Hence, by targeting FAP to DNA, we will eliminate confounding effects of damage elsewhere. (4) This photosensitizer tool requires far-red light, thus can be used for deep-tissue applications. By controlling illumination time, one can closely control ROS dosage and local production.

This new chemoptogenetic ROS generator can be targeted to proteins in various cellular locations (mitochondria, lysosome, etc.) and can also be targeted to other biomolecules and tissues to delineate the interplay of the hallmarks of aging. Using this novel chemoptogenetic tool, we anticipate to uncover novel causal mechanisms and their direct link to aging and lifespan. The FAP-chemoptogenetic tool, therefore, holds significant promise to address major questions in the field of DNA damage and aging.

## Figures and Tables

**Figure 1 genes-14-00485-f001:**
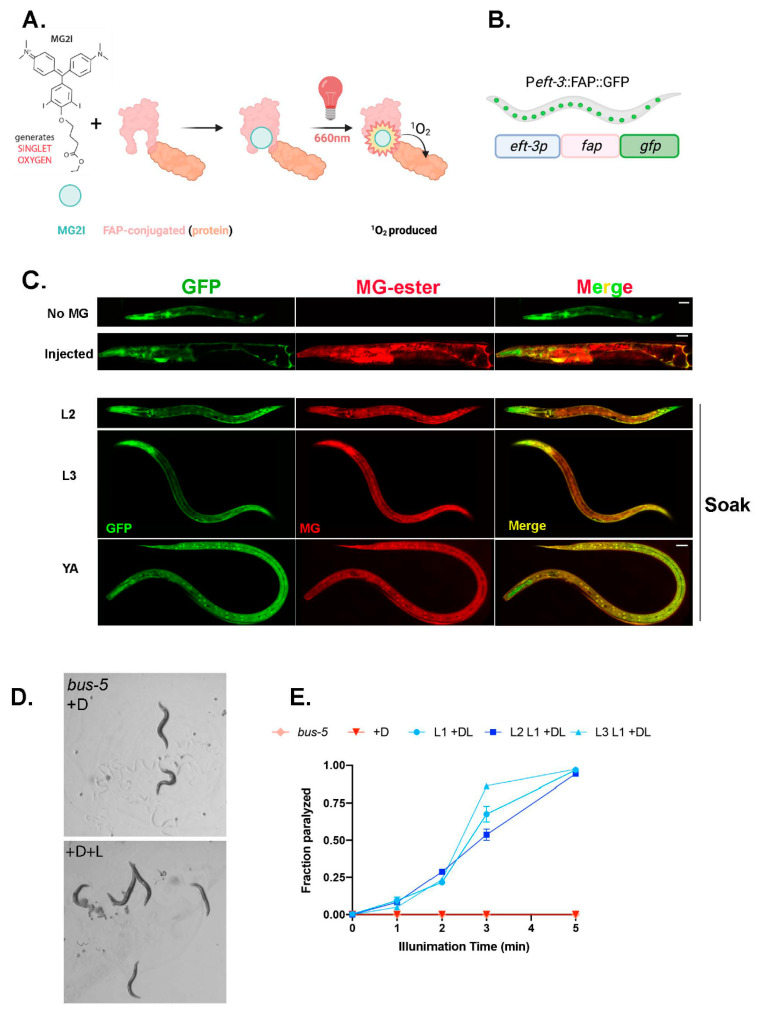
**MG-2I-chemoptogenetic tool** (**A**) Schematic of chemoptogenetic tool. MG-2I derivative generates ^1^O_2_ upon excitation with light. FAP can be tagged to any protein to target to specific cells or tissues. Once MG-2I binds to FAP and is illuminated with 660 nm far-red light, it generates singlet oxygen. Since ^1^O_2_ has limited permeability and is highly reactive, it will react in the immediate vicinity of its generation. (**B**) Schematic of *eft-3p::fap::gfp* in *bus-5* (**C**) Colocalization of GFP and MG-ester dye in transgenic *bus-5* worms. Top—no MG dye as negative control, D1 adults injected with MG-ester (200 μM) as positive control. Larvae (L2, L3) and young adults (YA) soaked in MG-ester (2 μM final concentration) for 24 hr. Scale bar is 25 μm. (**D**) Representative images of *bus-5* transgenic worms immediately after red light illumination. Light in the wavelength of 660 nm results in paralysis only in dye fed animals; dye only does not impact locomotion. (**E**) Paralysis rate quantified of 3 independent lines of *eft-3p::fap::gfp* in *bus-5,* and *bus-5* parental strain after 24 hr soak with MG-2I (2 μM final) and illuminated with red light (660 nm) for indicated time. A total of 3 independent experiments were performed for each line. Paralysis rates are plotted as mean ± SD. More than 30 worms were counted for each time point of each experiment.

**Figure 2 genes-14-00485-f002:**
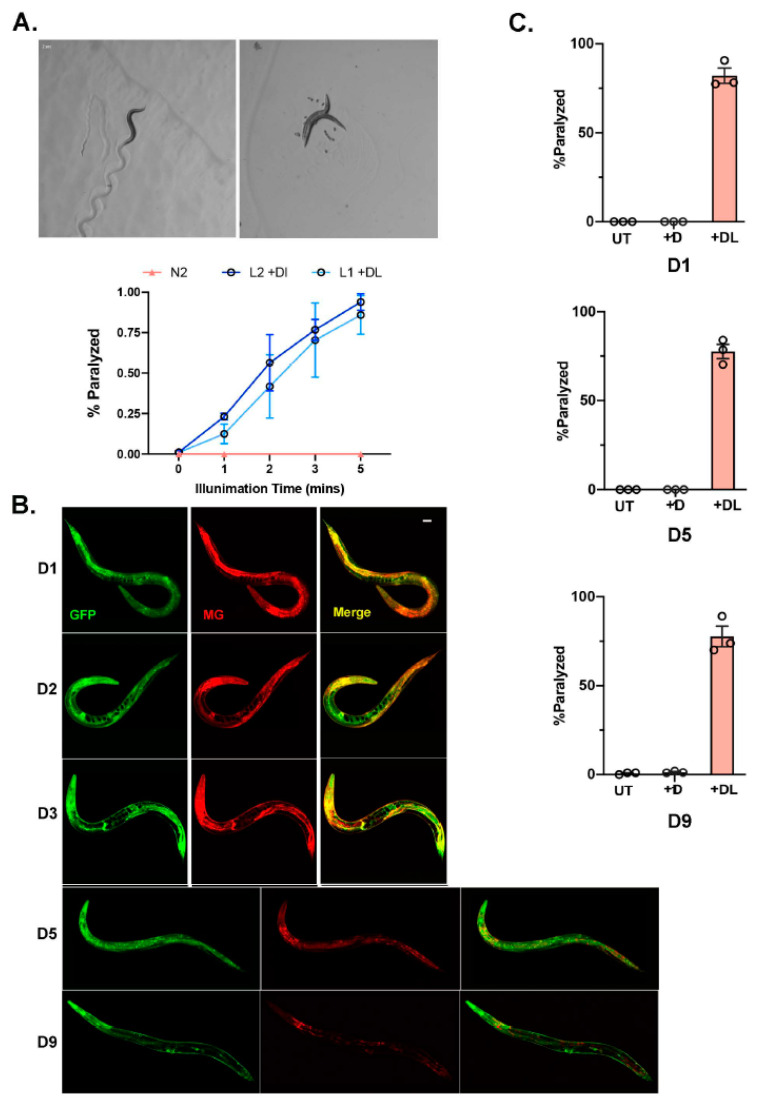
**Chemoptogenetic tool can be used to induce oxidative damage in a temporal manner** (**A**) N2 with *eft-3p::fap::gfp* as an extrachromosomal array was soaked in MG-2I overnight and illuminated with red light. Representative images of N2 integrated lines immediately after red light illumination. A total of 3 independent experiments were performed for each age group. Percent of paralyzed worms is plotted as mean ± SD. ~30 worms were counted for each experiment. (**B**) N2 with *eft-3p::fap::gfp* integrated lines were soaked with MG-ester as L4 for 24 hr. Colocalization of GFP and MG-ester dye was followed-up in day (D1), D2, D3, D5, and D9 adults. Scale bar = 25 μm. (**C**) Quantitation of paralysis immediately after red light illumination for 20 min in N2 with integrated *eft-3p::fap::gfp*. A total of 3 independent experiments were performed for each age group. Paralysis rate is plotted by mean ± SD. A total of 30–60 worms were counted for each experiment.

**Figure 3 genes-14-00485-f003:**
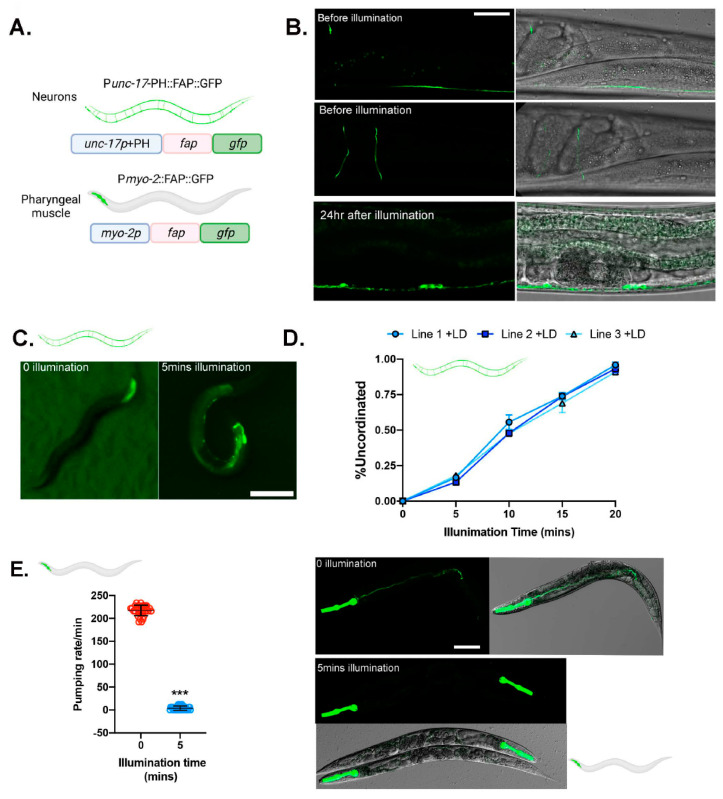
**Chemoptogenetic tool can be used to induce oxidative damage in a spatial manner** (**A**) Schematic of constructs targeting FAP to cholinergic motor neurons and pharyngeal muscle in *bus-5.* (**B**) Images of cholinergic motor neurons (GFP) before and 24 hr after illumination. Morphological changes, including absence of motor commissures and appearance of punctae, are visible after illumination. Scale bar = 25 μm (**C**) Representative images of *unc-17p::ph::fap::gfp* transgenic *bus-5* worms immediately after red light illumination. (**D**) Percentage of Unc animals after 24 hr MG-2I dye soak and immediately after red light illumination for the indicated times. (**E**) Pharyngeal pumping rate of *myo-2p::fap::gfp* in *bus-5*, fed OP50 was quantified before and after 24 hr MG-2I dye soak and immediately following 5 min illumination with far-red light. Pumping rate was measured in the terminal bulb. A total of 3 independent experiments were performed. Pumping rate was plotted as mean ± SD. Pumping rate was determined from the mean of 10–12 worms for each experiment. Unpaired *t*-test was performed. *** *p* < 0.001. (**F**) *myo-2p::fap::gfp* were fed OP50-GFP for 30 min, followed by 24 hr MG-2I dye soak and 5 min illumination. Brightly fluorescing *Escherichia coli (E. coli)* OP50-GFP bacteria observed in digestive tract towards the tail in non-illuminated worms, but absent after exposure to light. Scale bar = 90 μm.

**Figure 4 genes-14-00485-f004:**
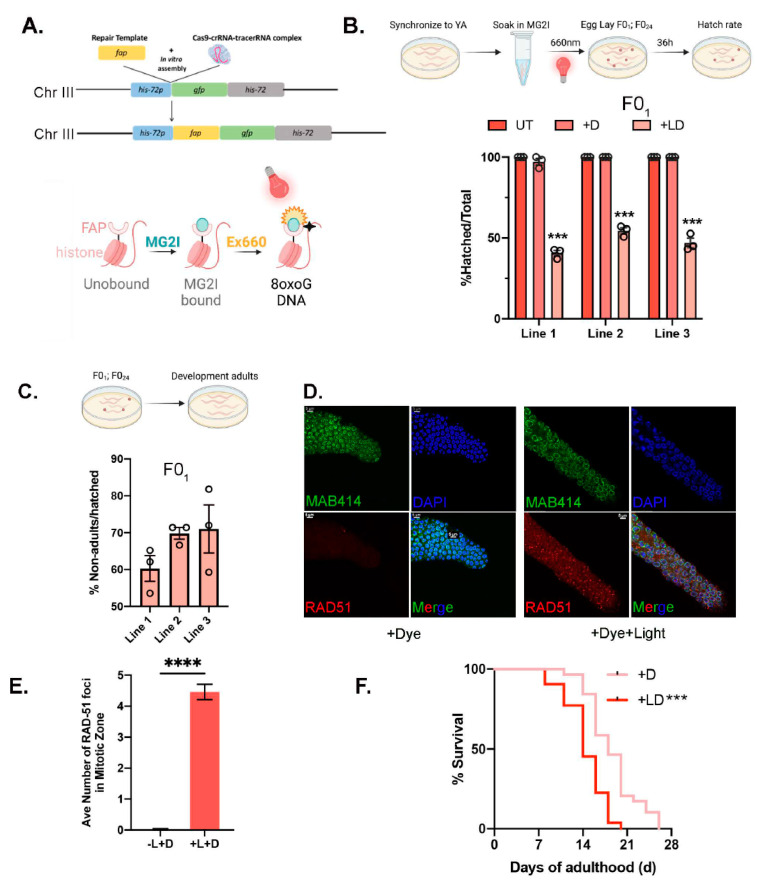
**FAP-induced oxidative DNA damage** (**A**) Schematic of CRISPR knock-in strategy to introduce FAP in HIS-72. crRNA was designed to target N terminal of *gfp::his-*72 of parental strain FAS46. CRISPR/Cas9-crRNA-tracerRNA complex was assembled in vitro and mixed with double stranded repair templates containing FAP coding sequence. Then the reaction mix was injected into gonads of adult FAS46 along with PRF4::*rol-6* injection marker. (**B**) Embryonic lethality (% unhatched eggs/total) was assessed for FAP-GFP-HIS-72 animals left untreated, soaked in MG-2I dye for 16 h (+D) and soaked in MG-2I and exposed to red light for 20 min. Data shown is for eggs collected 1 h after red light illumination (F0_1_); mean ± SEM. Four independent experiments performed: each experiment consists of 50 eggs laid by 20 worms. Experiment performed in triplicate at 20 °C. Two-way ANOVA with multiple comparisons. *** *p* < 0.001. (**C**) Development into young adults was determined in F0_1_. No developmental defects were noticed in +D or UT control lines. Percent hatched eggs that showed developmental defects is plotted as mean ± SEM. Four independent experiments performed: each experiment consists of ~50–70 eggs. Experiment performed in triplicate at 20 °C. (**D**) Germ lines from day 1 (D1) FAP-GFP-HIS-72 hermaphrodites treated with dye alone, or with dye + light fixed 2.5 h after illumination with red light and stained with DAPI (nuclei), MAB414 (membrane protein, control for staining) and RAD-51-antibody. Scale bar = 5 μm. (**E**) Percent of nuclei with indicated amount of RAD-51 foci plotted as mean ± SEM. Unpaired *t*-test were performed. *** *p* < 0.001. Two independent experiments were performed with foci in 47 and 85 nuclei in mitotic zone counted, respectively. (**F**) Representative survival curves of young adults soaked in MG-2I dye for 16 h (+D) and soaked in MG-2I and exposed to red light for 20 min. Survival curves were compared by Mantel–Cox log rank test. *** *p* < 0.001.

**Table 1 genes-14-00485-t001:** FAP expressing strains used in this study.

Targeting Tissue	Strains	Genotype/Transgene	Plasmids	Method
All	AUG4	*bus-5(br19) X*; *shEx1 [eft-3p::FAP::GFP::Unc-54 3′UTR]*	pSHH1	Extrachromosomal array with 20 ng/μL injection
All	AUG5	*bus-5(br19) X*; *shEx2 [eft-3p::FAP::GFP::Unc-54 3′UTR]*	pSHH1	Extrachromosomal array with 20 ng/μL injection
All	AUG6	*bus-5(br19) X*; *shEx3 [eft-3p::FAP::GFP::Unc-54 3′UTR]*	pSHH1	Extrachromosomal array with 20 ng/μL injection
All	AUG27	N2; *shIs1[eft-3p::FAP::GFP::Unc-54 3′UTR]*	pSHH1	Gamma irradiation
Cholinergic motor neuron	AUG24	*bus-5(br19) X*; *shEx4 [unc-17p::PH::FAP::GFP::Unc-54 3′UTR]*	pSHH2	Extrachromosomal array with 20 ng/μL injection
Cholinergic motor neuron	AUG25	*bus-5(br19) X*; *shEx5 [unc-17p::PH::FAP::GFP::Unc-54 3′UTR]*	pSHH2	Extrachromosomal array with 20 ng/μL injection
Cholinergic motor neuron	AUG26	*bus-5(br19) X*; *shEx6 [unc-17p::PH::FAP::GFP::Unc-54 3′UTR]*	pSHH2	Extrachromosomal array with 20 ng/μL injection
Pharynx	AUG28	*bus-5(br19) X*; *shEx7 [myo-2p:: FAP::GFP::Unc-54 3′UTR]*	pSHH3	Extrachromosomal array with 20 ng/μL injection
Pharynx	AUG29	*bus-5(br19) X*; *shEx7 [myo-2p:: FAP::GFP::Unc-54 3′UTR]*	pSHH3	Extrachromosomal array with 20 ng/μL injection
Pharynx	AUG30	*bus-5(br19) X*; *shEx7 [myo-2p:: FAP::GFP::Unc-54 3′UTR]*	pSHH3	Extrachromosomal array with 20 ng/μL injection
All	AUG31	*his-72* (*sh1* [FAP::GFP::HIS-72]) III		CRISPR

Plasmids were Gibson cloned, by inserting the target promoter amplified from N2 genome and *fap* sequence (amplified from [11]) into Fire vector, pPD95.75, which contains *gfp* and universal *unc-54 3′UTR.* The resulting plasmids were confirmed by sequencing and microinjected into gonad of adult N2 or *bus-5 (br19)* with 20 ng/μL concentration. Transgenic animals were isolated by GFP expression. N2; *shIs1[eft-3p::FAP::GFP::Unc-54 3′UTR]* was created by gamma irradiation following standard protocol [19].

## Data Availability

The data presented in this study are available in all figures (Figure 1, Figure 2, Figure 3 and Figure 4).

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
