# Peer review of "A Chemoptogenetic Tool for Spatiotemporal Induction of Oxidative DNA Lesions In Vivo"

_genes, 2023, doi:10.3390/genes14020485_

Round 1
Reviewer 1 Report
Han and colleagues describe a new tool in the nematode that would be of interest in the worm community. Interestingly, they describe a chemoptogenetic tool to control dosage and spatiotemporal induction of oxidative DNA damage. They have made key experiments to demonstrate these points in their strains, some of them, CRISPR-strains. They also check DNA damage by RAD51 inmunoflourescence.
I think that all the paper is correct in methods and procedures although I think the authors have to show the affirmation in line 208 and 209 “his-72::gfp or loss of his-72 did not affect lifespan in worms”. I think an extra control have to be added to say that, as some N2.
Author Response
We appreciate the reviewer's enthusiasm for the manuscript. We also thank the reviewer for critical feedback and have carefully revised the manuscript to reflect these changes (in the manuscript- please find with track changes).
The Reviewer correctly pointed out that we need to add a reference for his-72 loss/ his-72::GFP having no effect on lifespan. We have now added the following reference that has previously shown this- confirming our observation as well. (PMID: 27760329- Ref 30): Replication-Independent Histone Variant H3.3 Controls Animal Lifespan through the Regulation of Pro-longevity Transcriptional Programs
Reviewer 2 Report
Reviewed manuscript brings data confirming succesfull adaptation of chemoptogenic tool to induce ROS in live worms C. elegans. Presented results provide solid base of evidence of the temporal and cell-type specific effects of ROS induction by fluorogen activating peptide combined with iodinated malachite green (MG-2I) photosensitizer dye excitable with far red light. In addition, induction of oxidative DNA damage is addressed in the context of C.elegans developmental and lifespan impacts. Overall, the aim of the work is clearly defined, resulting in the study which may bring useful tool to the field. On the other hand, mansuscript is focused mainly on the presentation of the innovative method.
Points:
i) The efect of photosenzitizer used in the study is based on the formation of singlet oxygen. This radical has a very short half-life and reacts with the molecules in close vicinity, allowing highly localized reactions. Upon reaction with DNA (Ravanat et al., 2000) it primarily generates 8-oxoG. In the work of the authors, the source of ROS was localised either to the histones (reacting presumably with DNA) or to mebrane or to myosin structural protein. The question is, what is the most prominent ‘executive‘ oxidative damage that leads to cytotoxicity in the case of ROS localisation to mebrane and myosin (not proximal to DNA in both cases)? I would find beneficial, if this information is incorporated into the body of text.
ii) What may be the reason for different sensitivity to red light illumination (time in minutes) in Fig2C (upper panel, D1animals) vs Fig1E? To obtain the same rate of paralysis, 20 min red light (in Fig2C) vs 3 minutes (Fig1E) red light illumination was required. May it be the stability of reporter expression?
Author Response
We appreciate the reviewer's enthusiasm for the manuscript. We also thank the reviewer for critical feedback and have carefully revised the manuscript to reflect these changes (in the manuscript- please find track changes).
Point 1: Page 5 We have now included "To test the efficacy of neuron ablation with the FAP-chemoptogenetic system, we generated transgenic bus-5 worms that express FAP::GFP in cholinergic motor neurons, using the unc-17 promoter. Previous studies have reported target cell ablation can be achieved through localized ROS production27,28. Specifically, plasma membrane targeted ROS leads to lipid peroxidation at the membrane driving neuronal cell ablation and promoting paralysis28. Therefore, the N terminal of FAP::GFP was tagged with a membrane targeting sequence, the PH (Pleckstrin Homology) domain from rat PLC-δ29. PH domain binds to phosphoinositides and targets FAP to the plasma membrane. By targeting FAP to the plasma membrane we first wanted to test targeting singlet oxygen to lipids and lead to neuronal cell ablation." We have also added the pertinent references to the text.
Point 2: We apologize that we did not make this clear in the Results section. The Reviewer is correct: same rate of paralysis: 20 min red light (in Fig2C) vs 3 minutes (Fig1E) is because Fig 1E used extrachromosomal arrays (expressed in multiple copies) versus Fig 2 used integrated lines. We have now made this clear in the text "Since we expected eft-3p::fap::gfp integrated lines to retain low copy FAP, we used 20 min light exposure to examine paralysis."
As the reviewer pointed out, we have also carefully examined the manuscript for any grammar/ spelling checks. Thank you again.